# Evaluation of Mono- and Bi-Functional GLOBE-Based Vectors for Therapy of β-Thalassemia by *HBB^AS3^* Gene Addition and Mutation-Specific RNA Interference

**DOI:** 10.3390/cells12242848

**Published:** 2023-12-15

**Authors:** Lola Koniali, Christina Flouri, Markela I. Kostopoulou, Nikoletta Y. Papaioannou, Panayiota L. Papasavva, Basma Naiisseh, Coralea Stephanou, Anthi Demetriadou, Maria Sitarou, Soteroula Christou, Michael N. Antoniou, Marina Kleanthous, Petros Patsali, Carsten W. Lederer

**Affiliations:** 1Department of Molecular Genetics Thalassaemia, The Cyprus Institute of Neurology & Genetics, 6 Iroon Avenue, 2371 Nicosia, Cyprus; lkoniali@hotmail.com (L.K.); markella.costopoulou@hotmail.com (M.I.K.); nikolettap@cing.ac.cy (N.Y.P.); panayiotap@cing.ac.cy (P.L.P.); basman@cing.ac.cy (B.N.); coraleas@cing.ac.cy (C.S.); anthide@cing.ac.cy (A.D.); marinakl@cing.ac.cy (M.K.); 2Gene Expression and Therapy Group, Department of Medical and Molecular Genetics, King’s College London, Guy’s Hospital, London SE1 9RT, UK; christina.flouri@gmail.com (C.F.); michael.antoniou@kcl.ac.uk (M.N.A.); 3Thalassemia Clinic Larnaca, Larnaca General Hospital, 6301 Larnaca, Cyprus; msitarou@yahoo.gr; 4Thalassemia Clinic Nicosia, Archbishop Makarios III Hospital, 1474 Nicosia, Cyprus; chrnchr@spidernet.com.cy

**Keywords:** gene therapy, RNA interference, shRNAmiR, lentiviral vector, hemoglobinopathy, thalassemia, sickle cell anemia

## Abstract

Therapy via the gene addition of the anti-sickling β^AS3^-globin transgene is potentially curative for all β-hemoglobinopathies and therefore of particular clinical and commercial interest. This study investigates GLOBE-based lentiviral vectors (LVs) for β^AS3^-globin addition and evaluates strategies for an increased β-like globin expression without vector dose escalation. First, we report the development of a GLOBE-derived LV, GLV2-βAS3, which, compared to its parental vector, adds anti-sickling action and a transcription-enhancing 848-bp transcription terminator element, retains high vector titers and allows for superior β-like globin expression in primary patient-derived hematopoietic stem and progenitor cells (HSPCs). Second, prompted by our previous correction of *HBB^IVSI−110(G>A)^* thalassemia based on RNApol(III)-driven shRNAs in mono- and combination therapy, we analyzed a series of novel LVs for the RNApol(II)-driven constitutive or late-erythroid expression of *HBB^IVSI−110(G>A)^*-specific miRNA30-embedded shRNAs (shRNAmiR). This included bifunctional LVs, allowing for concurrent β^AS3^-globin expression. LVs were initially compared for their ability to achieve high β-like globin expression in *HBB^IVSI−110(G>A)^*-transgenic cells, before the evaluation of shortlisted candidate LVs in *HBB^IVSI−110(G>A)^*-homozygous HSPCs. The latter revealed that β-globin promoter-driven designs for monotherapy with *HBB^IVSI−110(G>A)^*-specific shRNAmiRs only marginally increased β-globin levels compared to untransduced cells, whereas bifunctional LVs combining miR30-shRNA with β^AS3^-globin expression showed disease correction similar to that achieved by the parental GLV2-βAS3 vector. Our results establish the feasibility of high titers for LVs containing the full *HBB* transcription terminator, emphasize the importance of the *HBB* terminator for the high-level expression of *HBB*-like transgenes, qualify the therapeutic utility of late-erythroid *HBB^IVSI−110(G>A)^*-specific miR30-shRNA expression and highlight the exceptional potential of GLV2-βAS3 for the treatment of severe β-hemoglobinopathies.

## 1. Introduction

Hemoglobinopathies are among the most common monogenic disorders, are almost universally of recessive inheritance, and cause clinical symptoms when affecting the α- and β-globin chains as constituents of the main adult hemoglobin, HbA (α_2_β_2_) [1]. Hemoglobinopathies may be brought about by toxic protein variants, such as by the sickling β-globin E6V amino acid change (β^S^), or by the reduction or elimination of α- and β-globin expression in α- and β-thalassemia, respectively. Sickle cell disease and β-thalassemia may both reach extreme severity from infancy onwards, and without adequate management via blood transfusion and iron chelation, they manifest as potentially lethal hemolytic anemias. Originally mostly confined to malaria regions, where a carrier status for hemoglobinopathies conferred a selective advantage through resistance to *Plasmodium falciparum*, hemoglobinopathies, though considered rare diseases, are now widespread through carrier migration and represent a global health challenge [2]. The last four decades have therefore seen tremendous efforts to develop advanced therapies for their treatment.

The lentiviral vector (LV) transfer of β-globin (*HBB*) or *HBB*-like globin genes into hematopoietic stem and progenitor cells (HSPCs) represents a promising curative therapeutic approach for transfusion-dependent β-thalassemia patients who lack a suitable human leukocyte antigen (HLA)-matched sibling donor [3,4,5,6]. However, results from ongoing clinical trials indicate a genotype dependence of the clinical benefits and achievement of transfusion independence [7,8,9,10,11]. For one of the most common β-thalassemia mutations in Mediterranean populations, the *HBB^IVSI−110(G>A)^* (*HBB*: c.93-21G>A, rs35004220), an aberrant splice acceptor site 19 bp 5′ to the normal acceptor of intron I (IVSI), results in the generation of aberrant *HBB* mRNA (*HBB^IVSI−110(G>A)^* mRNA) with an integrated 19-bp intronic segment and an in-frame premature stop codon that is preferentially used by splicing machinery [12,13]. Despite marked normal *HBB* pre-mRNA splicing and the presumed reduction of the aberrant transcript by nonsense-mediated mRNA decay (NMD), *HBB^IVSI−110(G>A)^* mRNA remains prevalent in the erythroid cells of affected homozygous patients and disease models [14], causing defective *HBB* expression and markedly reduced HbA synthesis [15,16]. Importantly, and in line with preclinical studies indicating that the presence of the *HBB^IVSI−110(G>A)^* allele interferes with the production of vector-derived HBB, results from clinical trials show patients with the *HBB^IVSI−110(G>A)^* mutation to be recalcitrant to LV-mediated *HBB* gene transfer compared to other *HBB* mutations with residual HBB expression (β^+^ mutations) [4,17,18].

Thus far, strategies attempting to suppress abnormal *HBB^IVSI−110(G>A)^* mRNA expression or promoting use of the normal IVSI 3′ splice acceptor site have been shown to restore normal HBB production and may therefore greatly improve the efficacy of therapy through gene addition in *HBB^IVSI−110(G>A)^* β-thalassemia patients [19,20,21,22]. In a proof-of-concept study by our group, potent shRNAs selectively targeting the aberrant *HBB^IVSI−110(G>A)^* mRNA effectively restored physiological HBB expression and provided evidence that RNA interference (RNAi) alone or combined with the delivery of a functional *HBB* transgene would provide clinical benefits. However, we also demonstrated the toxicity of control and test shRNAs in CD34^+^ cells, likely due to the employment of constitutive high-level U6-promoter-driven RNA-polymerase (III) (RNApol(III))-based shRNA expression and the corresponding saturation of the endogenous RNAi machinery [22]. Recent publications have highlighted the design criteria and potential efficacy of same-vector combined transgene and RNApol(II)-based shRNAmiR expression, such as for the silencing of α-globin or of the sickling mutation concurrent with the overexpression of a β-like globin transgene. In the process, RNApol(II)-driven miRNA scaffolds have been studied extensively and been shown to reduce nonspecific toxicities related to shRNA overexpression [23,24,25,26]. Mutation-specific LV enhancement for *HBB^IVSI−110(G>A)^*-affected individuals may therefore be achieved by the expression of miRNA-embedded short hairpin RNA (shRNAmiR) sequences targeting *HBB^IVSI−110(G>A)^* mRNA with a view to reducing the potential interference of aberrant mRNA and to enhancing incorporation of endogenous HBB or of vector-derived HBB^AS3^ into hemoglobin tetramers.

In addition to mutation-specific enhancements, we also considered the existing scope for the improved transgene expression and efficacy of gene addition-based therapies without increasing the vector load and potential genotoxicity risks [25]. In this context, we considered findings by Proudfoot and colleagues potentially important, who demonstrated that presence of a transcriptional terminator element 0.9–1.6 kb downstream of the *HBB* polyadenylation site (poly(A)) is associated with higher mature *HBB* mRNA and protein levels [27,28], which is in part achieved by promoting more efficient mRNA 3′ end formation and reducing the susceptibility of transcripts to degradation. An additional consideration was that GLOBE-based LVs expressing the triple amino acid substitution anti-sickling human β-globin transgene (*HBB^AS3^*, β^AS3^) would potentially be therapeutic for both major hemoglobinopathies: β-thalassemia and sickle cell diseases. Moreover, LVs bearing *HBB^AS3^* under the transcriptional control of a short β-globin promoter and the key HS2 and HS3 β-globin locus control region (β-LCR) elements have already demonstrated efficacy in the restoration of β-/α-globin chain balance and the amelioration of the β-hemoglobinopathy phenotype in vitro and in animal models [29,30,31], as well as in ongoing clinical trials (NCT02247843, NCT03964792) in sickle cell disease and β-thalassemia patients [32,33].

Accordingly, this study aimed to pursue two incremental strategies to enhance the therapeutic potential of the well-characterized *HBB*-encoding GLOBE LV [5,34] in general and to improve its performance for *HBB^IVSI−110(G>A)^* in particular. With our assessment criterion being the ability to induce hemoglobin production and correct *HBB^IVSI−110(G>A)^* β-thalassemia in vitro, we tested the triple-anti-sickling GLOBE derivative GLV1-βAS3 as base vector and found that the insertion of an additional 848-bp transcription termination element, resulting in GLV2-βAS3, displayed superior β^AS3^ expression efficiency in β-thalassemia patient-derived HSPCs. Based on GLV2-βAS3 and control designs, our experiments further showed that erythroid-specific RNApol(II)-driven designs for the isolated expression of *HBB^IVSI−110(G>A)^*-specific shRNAmiR only marginally increased HBB levels compared to untransduced cells. Likewise, for moderate vector copy numbers (VCNs), bifunctional LVs combining shRNAmiR|β^AS3^-globin expression were comparable to the parental GLV2-βAS3 vector for their improvement of β-/α-globin ratios in *HBB^IVSI−110(G>A)^*-homozygous HSPCs. Overall, our results highlight the therapeutic potential of the GLV2-βAS3 LV and emphasize the importance of the *HBB* transcription terminator element for the high-level expression of *HBB*-like transgenes and for the treatment of patients with more severe β-thalassemia mutations.

## 2. Materials and Methods

### 2.1. Design and Construction of miR30^shRNA^-Expressing LVs

The gene addition LVs, GLV1-βAS3 and GLV2-βAS3, derivatives of the self-inactivating GLOBE vector [34], provided the backbone for the generation of shRNAmiR-bearing human cytomegalovirus (CMV) promoter-driven vectors encoding the triple-amino acid substitution anti-sickling *HBB^AS3^* transgene (*HBB^G16D,E22A,T87Q^*, β^AS3^) [35]. The *HBB^AS3^* cassette is transcribed in antisense orientation relative to the 5′ long terminal repeat CMV promoter and is under the control of a short (264-bp) promoter and key HS2 and HS3 elements of the βLCR. For GLV2-βAS3-based constructs, an additional 848-bp sequence containing 809 bp of *HBB* transcription termination (β-Term) sequence (genomic coordinates [hg38], chr11: 5 224 639-5 223 832, see Appendix A) was inserted at the *HpaI* restriction site of the GLOBE LV for more efficient processing of the pre-mRNA. For the generation of shRNAmiR-expressing LVs, the miR30 artificial miRNA-expression strategy described by Du and colleagues was employed, in which each shRNA^miR30^ sequence was flanked by *Age*I and *Afe*I sites and featured two inverted BsmBI sites separating the 125-nt 5′ and 3′ miR30 arms from the internal hairpin sequence [36]. The GLV2-miR30shRNA vectors were generated by replacing the whole *HBB^AS3^* transgene with the shRNA^miR30^-based expression cassette (Construct 1—Appendix A) using *Cla*I and *Swa*I restriction sites. For the generation of the GLV2-βAS3-miR30shRNA vector, the miR30-expression cassette was cloned between the *Age*I and *Afe*I sites of the IVSII (positions c.303–163) of the *HBB^AS3^* transgene (Construct 2—Appendix A). Oligonucleotides representing the target-specific 21-nt sense and antisense strands of shRNAs linked by a 19-nt loop structure were inserted in the two inverted BsmBI sites, including the *HBB^IVSI−110(G>A)^*-targeting shMID, shMIDA and shMIDB. Control constructs consisted of a scrambled shSCR and a GFP-targeting shGFP. All oligonucleotides used (see Appendix A) were obtained from GenScript (Piscataway, NJ, USA), and all plasmids were constructed via standard procedures and sequence-verified before viral production.

### 2.2. Cell Lines, Culture and Differentiation

Murine erythroleukemia (MEL) cells of the APRT^−^ cell line C88 [37], originally described by Deisseroth and Hendrick [38], MEL derivatives expressing green fluorescent protein (MEL-GFP) [39] and the human-*HBB^IVSI−110(G>A)^*–transgenic (MEL-*HBB^IVS^*) cells [22] mimicking the *HBB^IVSI−110(G>A)^* splice defect, were used in the current study. MEL and derived cell lines were cultured in RPMI, and human embryonic kidney (HEK) 293T cells in IMDM, both supplemented with 10% fetal bovine serum (FBS), 100 U/mL penicillin/100 g/mL streptomycin (1x penicillin/streptomycin) and 100 mM L-glutamine (all Invitrogen, Thermo Fisher Scientific, Waltham, MA, USA), in a humidified atmosphere at 37 °C and 5% CO_2_.

For functional analysis of the LVs, cells were transduced with hourly agitation for six hours in polybrene-containing medium as previously described [39]. MEL, MEL-*HBB^IVS^* and MEL-GFP cells were induced to undergo erythroid differentiation by culturing an initial concentration of 2 × 10^5^ cells/mL for 10 days in medium containing 1.5% DMSO (Sigma-Aldrich, Irvine, UK). Cells were collected on day 4 for RNA and day 7–10 post-induction of erythropoiesis for protein analysis.

### 2.3. LV Production and Titration

All lentiviral stocks were produced by the calcium phosphate-mediated transient transfection of HEK293T cells, as previously published [40]. In brief, HEK293T cells were seeded onto 100-mm dishes and allowed to grow overnight to 80% confluency. The media were replaced prior to transfection, and LVs were produced by co-transfecting 12.8 μg lentiviral transfer vector plasmids, pCMVΔ8.74 (2nd-generation gag/pol packaging construct), 2.5 μg of pRSV-REV (3rd-generation REV construct), 6 μg of pAdVantage (encoding viral translation initiation enhancers for higher protein expression) and 3.6 μg of pMD2.VSVG (envelope plasmid) into cells. At 48 h post-transfection, viral supernatants were filtered through 0.45 μm Durapore filters (Millipore, Bedford, MA, USA) and concentrated 350-fold by 4 h centrifugation at 20,000 RCF and 4 °C. Viral stock titers were determined by the transduction of MEL cells with serial dilutions of viral preparation and the quantification of VCN/cell via quantitative PCR (qPCR) 14 days post-transduction, as previously described [40].

### 2.4. Human CD34^+^ Cell Isolation, LV Transduction and Differentiation

All human CD34^+^ cell samples were obtained from *HBB^IVSI−110(G>A)^*-homozygous patients attending the Thalassemia Clinics, Cyprus, as part of their routine clinical care and based on written informed consent. Up to 10 mL of venous blood was obtained under aseptic conditions and CD34^+^ cells were isolated from mononuclear cells using Accu-Prep Lymphocytes (Axis-Shield PoC AS, Dundee, Ireland) and a CD34 MicroBead Kit (Miltenyi Biotec, Bergisch Gladbach, Germany) as previously published [39]. The isolated cells were cryopreserved in liquid nitrogen (50% fetal bovine serum, 40% culture medium and 10% dimethyl sulfoxide) or immediately used for LV transduction.

Human CD34^+^ cells were cultured in StemSpan^TM^ SFEM II medium (Stemcell Technologies, Vancouver, BC, Canada) supplemented with 1x CC100 (Stemcell Technologies, Vancouver, BC, Canada), 2 unit/mL erythropoietin (Binocrit; 4 000 IU/0.4 mL, Sandoz GmbH, Kundl, Austria), 10^−6^ M dexamethasome (Sigma-Aldrich, Munich, Germany) and 1x penicillin/streptomycin (Thermo Fisher Scientific, Waltham, MA, USA). At 48 h post-transduction, cells were transferred to a differentiation medium consisting of 70% Minimum Essential Medium Eagle, Alpha modification (Sigma-Aldrich, Munich, Germany), 30% defined FBS (Hyclone GE Healthcare, Logan, UT, USA), 10 μM 2-mercaptoethanol (Sigma-Aldrich GmbH, Munich, Germany), 10 U/mL stem cell factor (PeproTech, Rocky Hill, CT, USA) and 1x penicillin/streptomycin (Thermo Fisher Scientific, Waltham, MA, USA).

### 2.5. RNA Extraction and RT-qPCR

On day 4 of differentiation, total RNA was isolated from 0.5 × 10^6^ cultured cells using Trizol reagent (Invitrogen, Thermo Fisher Scientific, Waltham, MA, USA) according to the manufacturer’s instructions. A 500-ng aliquot of DNase-I-treated RNA (Invitrogen™, Thermo Fisher Scientific, Waltham, MA, USA) was used for complementary DNA (cDNA) synthesis using the TaqMan Reverse Transcription PCR kit (Applied Biosystems, Thermo Fisher Scientific, Waltham, MA, USA) following the manufacturer’s instructions. The resulting cDNA samples were diluted four-fold with RNase-free water (Sigma-Aldrich, Munich, Germany) to a final cDNA concentration equivalent to 12.5 ng/µL total RNA, and 2 µL was used for each qPCR reaction. Relative quantification of gene expression was performed using the SYBR Green qPCR amplification detection system on the 7900HT Fast Real-Time PCR System (both Applied Biosystems, Foster City, CA, USA). Appendix A provides a list of the primers and probes used in the study. The levels of aberrantly and normally spliced *HBB* mRNA were quantitatively determined by qPCR using the conditions previously described [14].

### 2.6. Immunoblots

Cell lysates equivalent to 10 μg total protein were resolved on 12% polyacrylamide gels and blotted onto 0.4 μm nitrocellulose Parablot NCP membrane (Macherey-Nagel GmbH, Düren, Germany) using wet electrophoretic transfer. Membranes were blocked in 5% BSA (Roche, Basel, Switzerland) and incubated overnight at 4 °C with the appropriate primary antibody: mouse-anti-HBB (sc-21757; 1:1000), rabbit-anti-mHba (sc-21005; 1:1000) (all Santa Cruz Biotechnologies, Dallas, TX, USA), GFP tag Polyclonal antibody (50430-2-AP, 1:5000; Proteintech, Planegg-Martinsried, Germany), or mouse-anti-mActb (A1978, 1:10,000; Sigma-Aldrich, Munich, Germany). Antibody-conjugated proteins were detected through the incubation of membranes with 1:10,000 diluted anti-mouse or anti-rabbit horseradish peroxidase-conjugated secondary antibodies (Dako/Agilent, Santa Clara, CA, USA) and visualization was accomplished using the Clarity Western ECL Substrate (Lumisensor, GenScript, Piscataway, NJ, USA) according to the manufacturer’s instructions, on the UVP Biospectrum 810 Imaging system (Thermo Fisher Scientific, Waltham, MA, USA). Densitometry analysis was performed using the NIH ImageJ software (v1.53p, National Institute of Health, Bethesda, MD, USA).

### 2.7. High-Performance Liquid Chromatography (HPLC)

Globin chains and hemoglobins in primary erythroid cells were determined by reverse-phase-HPLC (RP-HPLC) and cation-exchange HPLC (CE-HPLC), respectively, on the last day of differentiation, as previously described [26,41]. Primary erythroid pellets of 10^6^ cells were lysed in HPLC-grade water supplemented with 5 mM 1,4-dithiothreitol (DTT; Thermo Fisher Scientific, Waltham, MA, USA), centrifuged at 21 100 RCF for 10 min at 4 °C, and the equivalent of 2–5 × 10^5^ cells (1–25 μL of the supernatant) was injected for analysis on a LC-20AD chromatographic system (Shimadzu, Kyoto, Kyoto, Japan). The Aeris Widepore C18 column (Phenomenex, Torrance, CA, USA) was used for RP-HPLC to separate peptides based on their hydrophobicity using a linear gradient of acetonitrile/0.1% trifluoroacetic acid against 0.1% trifluoroacetic acid/0.033% sodium hydroxide. A PolyCAT A™ Column (PolyLC Inc., Columbia, MD, USA) was used for CE-HPLC to separate the hemoglobins based on their isoelectric point with a rising linear gradient of 40 mM Bis-tris/2 mM KCN/200 mM NaCl (pH 6.8) against 40 mM Bis-tris/2 mM KCN/200 mM NaCl (pH 6.5) (all Merck, Darmstadt, Germany). Heme and globin chains for RP-HPLC and hemoglobins for CE-HPLC were identified as absorbance peaks at 190 nm and 417 nm, respectively, by their characteristic elution times as determined using cord blood and commercial HbA0 and HbA2 preparations (both Sigma-Aldrich/Merck KGaA, Darmstadt, Germany) as controls. Areas under peaks were used to determine the relative quantities of the globin chains and hemoglobins in samples.

### 2.8. Cytocentrifugation and Microscopy

Morphological analysis of transduced cells during erythropoiesis was performed after cytocentrifugation and the May–Grünwald–Giemsa (Fluka, Munich, Germany) and dianisidine (Sigma-Aldrich, Munich, Germany) staining of cells spotted on slides. Representative images of the cytocentrifuge preparation of early-stage and terminally differentiated cells were imaged, using light microscopy, on an IX73P1F inverted microscope with CellSens 1.7 imaging software (all Olympus Corporation, Shinjuku City, Tokyo, Japan).

### 2.9. Flow Cytometry

Cell viability and GFP levels were assessed via flow cytometry. For the cell viability assessment, cells were washed in PBS and stained with propidium iodide and YO-PRO^TM^-1 Iodide (491/509) reagents (both Thermo Fisher Scientific, Waltham, MA, USA) at the indicated time points of cell differentiation. To assess the knockdown efficiency of GLV2-miR30shRNA, GLV2-βAS3-miR30shRNA and Ef1a-miR30shRNA LVs, mock-transduced MEL and LV-transduced MEL-GFP cells were washed with PBS and assessed for eGFP expression at the indicated time points of differentiation. Data were acquired on a CyFlow Cube 8 6-channel instrument (Sysmex Partec, Münster, Germany) and analyses were performed with the FCS Express 7 software (DeNovo Software, Glendale, CA, USA).

### 2.10. Statistical Analysis

Statistical analysis was performed using the GraphPad Prism version 7.0 software (GraphPad Software Inc., La Jolla, CA, USA). Bar charts show arithmetic means ± standard deviation of two to three independent experiments, as indicated. Groupwise tests compared samples with mock-treated controls, unless indicated otherwise, and were performed as detailed in the legend entry for each individual panel.

## 3. Results

### 3.1. Design and Characterization of GLOBE-Based HBB^AS3^ Transgene-Expressing LVs with a Transcription Termination Sequence for Enhanced Expression

Mindful of the potential utility of transcriptional terminator sequences and β^AS3^-based LV designs for effective therapy of β-hemoglobinopathies by gene addition, we set out to evaluate whether the addition of an *HBB* transcription termination sequence could enhance *HBB^AS3^* transgene expression and improve the therapeutic efficacy of GLOBE-based gene addition LVs. The GLV1-βAS3 derivative of the GLOBE LV, harboring the *HBB^AS3^* transgene and βLCR, was modified by inserting an 848-bp sequence encompassing the β-Term region downstream of the *HBB* poly(A) site, and was designated GLV2-βAS3 (Figure 1A).

GLV1-βAS3 and GLV2-βAS3 LVs were produced by the standard calcium phosphate-mediated transient transfection of HEK293T cells and concentrated by centrifugation. The functional titers of the two LVs were determined by transducing MEL cells and measuring the average number of vector copies integrated per genome via qPCR on day 14 post-transduction. The ranges of the infectious titers for both GLV1-βAS3 and GLV2-βAS3 across all production batches were similar, at 2.32 × 10^9^ ± 1.85 × 10^9^ for GLV1-βAS3 and 1.47 × 10^9^ ± 1.2 × 10^9^ for GLV2-βAS3, leading us to conclude that the insertion of the β-Term does not impact the viral particle’s production and infectivity (Figure 1B).

In order to evaluate whether the additional β-Term sequence enhances *HBB^AS3^* transgene expression in cells of the erythroid lineage, MEL cells were transduced with an increasing number of infectious particles (MOI 0.5, 1 and 2) and differentiated in DMSO-containing culture medium 14 days post-transduction (Appendix A). Vector-derived *HBB^AS3^* mRNA expression, determined via RT-qPCR on day 4 of differentiation, revealed that the presence of the β-Term element in GLV2-βAS3-transduced cells significantly enhanced *HBB* expression compared to GLV1-βAS3-transduced cells (Appendix A). *HBB^AS3^* mRNA expression corrected for VCNs for both LVs revealed a 5.0-fold (MOI-0.5), 3.8-fold (MOI-1, *p* < 0.01) and 9.1-fold (MOI-2, *p* < 0.05) increase in vector-derived *HBB^AS3^* transgene expression in GLV2-βAS3- compared to GLV1-βAS3-transduced cells. Similarly, the analysis of the effect of the β-Term element on the levels of HBB protein revealed a 2.9 (MOI-0.5), 2.9 (MOI-1, *p* < 0.01) and 3.9 (MOI-2, *p* < 0.0001)-fold higher level of HBB chain production in GLV2-βAS3-transduced cells compared to GLV1-βAS3 (Appendix A). Collectively, these data highlight the superior transgene expression at both the mRNA and protein level for the GLV2-βAS3 over the GLV1-βAS3 vector.

The therapeutic potential of GLV1-βAS3 and GLV2-βAS3 in terms of transgene expression and correction of the β-thalassemia phenotype at a therapeutically relevant MOI (MOI = 3), was further assessed in CD34^+^ HSPCs derived from the peripheral blood of *HBB^IVSI−110(G>A)^* β-thalassemia homozygous individuals (Figure 1C). Starting 48 h post-transduction, the cells were differentiated for 9 days, without LV transduction affecting their viability or erythroid proliferation. Quantification of the vector-derived *HBB^AS3^* transgene expression via RT-qPCR, and the normalization of results for VCN/cell to adjust for variable gene transfer, revealed a more than 1.9-fold expression in GLV2-βAS3- compared to GLV1-βAS3-transduced cells at equivalent VCNs, whereas endogenous *HBB* mRNA levels remained unaffected (Figure 1D,E). This was associated with greatly reduced aberrant *HBB^IVSI−110(G>A)^* mRNA for GLV2-βAS3 compared to GLV1-βAS3 transduction (Figure 1F).

Importantly, immunoblot analysis likewise confirmed a significantly higher increase in vector-derived HBB^AS3^ protein levels in GLV2-βAS3- over GLV1-βAS3-transduced cells (Figure 1G,H). CE-HPLC and RP-HPLC analyses were used to assess the relative production of transgene-derived HbA^βAS3^ and endogene-derived HbA levels (Figure 1I) and the HBB/HBA ratios (Figure 1J), respectively, in terminally differentiated erythroid cells. This analysis revealed that GLV2-βAS3-transduced cells were associated with 2-fold vector-derived HbA^βAS3^ compared to GLV1-βAS3-transduced cells at the equivalent VCN/cell (Figure 1K). This was concomitant with the morphological correction of the β-thalassemia phenotype in primary cells from β-thalassemia individuals for both vectors (Figure 1L).

Taken together, these data confirm the superior *HBB^AS3^* expression efficiency of GLV2-βAS3 in HSPCs compared to the parental GLV1-βAS3 vector.

### 3.2. Design and Functional Validation of LVs Expressing shRNAmiR under the HBB Promoter Alone (GLV2-shRNAmiR) or as Part of IVSII of the HBB^AS3^ Transgene (GLV2-βAS3-shRNAmiR)

We hypothesized that the silencing of the aberrantly spliced *HBB^IVSI−110(G>A)^* mRNA in *HBB^IVSI−110(G>A)^* β-thalassemia would reduce its competition with other transcripts, and therefore promote its expression and contribution to hemoglobin formation from endogenous HBB as well as vector-derived HBB^AS3^. Therefore, the GLV2-βAS3 vector was modified so as to allow the expression of a miRNA-30-based shRNA expression cassette from the *HBB* promoter either alone or in combination with the *HBB^AS3^* transgene. Thus, two sets of GLOBE-based LVs encoding shRNAmiR were generated. First, LVs were constructed to evaluate individually the therapeutic potential of the different shRNAmiR designs. Second, LVs were also built to assess whether shRNAmiR specifically targeting the aberrant *HBB^IVSI−110(G>A)^* mRNA could be used as a monotherapy for the *HBB^IVSI−110(G>A)^* β-thalassemia. Specifically, LVs encoding the *HBB* promoter-driven miR30-based shRNA-expression cassette in place of the *HBB^AS3^* transgene were generated and denoted GLV2-shRNAmiR, specifically the vectors GLV2-shMIDmiR, GLV2-shMIDAmiR, GLV2-shMIDBmiR, GLV2-shSCRmiR and GLV2-shGFPmiR. In parallel, in order to evaluate whether the shRNAmiR-mediated knockdown of the aberrant *HBB^IVSI−110(G>A)^* mRNA could be used to enhance the therapeutic efficacy of GLOBE-based *HBB* addition LVs, bifunctional vectors were generated, in which the miR30-based shRNA expression cassette was inserted into the IVSII of the *HBB^AS3^* transgene, thus allowing for combined expression of the *HBB^AS3^* transgene and knockdown of the aberrant *HBB^IVSI−110(G>A)^* mRNA. These bifunctional vectors were designated GLV2-βAS3-shRNAmiR, and, specifically, GLV2-βAS3-shMIDmiR, GLV2-βAS3-shMIDAmiR, GLV2-βAS3-shMIDBmiR, GLV2-βAS3-shSCRmiR and GLV2-βAS3-shGFPmiR (Figure 2A). Conceptionally, such bifunctional LVs could result in full disease correction at lower VCNs than either the *HBB* gene addition or RNAi strategies alone, and thus enable safer therapy.

Reports of the lowered efficiency of shRNAmiR expression and knockdown from RNApol(II)-based promoters compared to constitutively active RNApol(III)-based promoters prompted us to evaluate constitutive shRNA expression as a direct comparison of therapeutic efficacies [42,43]. To this end, one of our published U6-driven LVs expressing a potent shRNA targeting the aberrant *HBB^IVSI−110(G>A)^* mRNA (LV-U6-shMID) and a respective scrambled shRNA sequence as the negative control (LV-U6-shSCR) were included in our analyses, which also served as a reference point for our previous work [22]. Additionally, our novel miR30-based shRNA expression cassette was also inserted into a constitutively active RNApol(II)-based *Ef1a* promoter-driven expression vector to generate LV-Ef1a-shRNAmiR vectors, specifically LV-Ef1a-shMIDmiR, LV-Ef1a-shMIDAmiR, LV-Ef1a-shMIDBmiR, LV-Ef1a-shSCRmiR and LV-Ef1a-shGFPmiR (Appendix A), to allow analysis of the effect of differing spatiotemporal RNApol(II)-driven expressions on phenotype correction.

Functional titers of concentrated LV preparations for all vectors were measured on MEL cells, and no significant difference in the functional vector titers was observed between the parental, the GLV2-shRNAmiR and the GLV2-βAS3-shRNAmiR set of LVs across all vector productions (Figure 2B). This suggested that any impact of the miR3.0-expression cassette insertions into IVSII of the *HBB^AS3^* transgene on viral particle production and infectivity would be minor.

### 3.3. Functional Assessment of Monofunctional GLV2-miR30shRNA and Bifunctional GLV2-βAS3-miR30shRNA LVs in MEL-GFP Cells

The functionality of β-globin promoter-driven GLV2-shRNAmiR and bifunctional GLV2-βAS3-shRNAmiR vectors and the efficacy of shRNAmiRs in suppressing the target transcript at low copy numbers (VCN 3), were first evaluated and compared to those of the constitutively active *Ef1a* promoter-driven vectors. MEL-GFP cells, stably expressing eGFP from the *PGK* promoter, were transduced with LVs bearing shGFPmiR (LV-Ef1a-shGFPmiR, GLV2-shGFPmiR and GLV2-βAS3-shGFPmiR) or the respective scrambled shSCRmiR control LVs LV-Ef1a-shSCRmiR, GLV2-shSCRmiR and GLV2-βAS3-shSCRmiR, and differentiated in DMSO-containing culture medium for 14 days post-transduction (Figure 2C).

Messenger RNA expression levels for eGFP and for vector-derived miR30 and *HBB^AS3^* transcripts were assessed by RT-qPCR in LV- and mock-transduced MEL-GFP and mock-transduced MEL cells, collected on day 4 of differentiation (Figure 2D–F). After normalization for VCN, a similar pattern of reduction in the eGFP mRNA levels was observed across all bulk populations of the MEL-GFP cells transduced with the different shGFPmiR-expressing vectors over their respective shSCRmiR controls (50% in GLV2-shGFPmiR over GLV2-shSCRmiR, 53% in GLV2-βAS3-shGFPmiR over GLV2-βAS3-shSCRmiR-transduced cells and 37% in LV-Ef1a-shGFPmiR over LV-Ef1a-shSCRmiR). This indicated the equivalent silencing of GFP by the erythroid-specific βLCR-*HBB* promoter-driven monofunctional GLV2-shGFPmiR and bifunctional GLV2-βAS3-shGFPmiR vectors and by the constitutively active *Ef1a*-promoter-driven LV-Ef1a-shGFPmiR vector (Figure 2D). Of note, expression of the vector-derived *HBB^AS3^* transgene was comparable between cells transduced with the parental GLV2-βAS3 and the bifunctional GLV2-βAS3-shGFPmiR/GLV2-βAS3-shSCRmiR vectors. This demonstrates that neither the insertion of the miR-30-based shRNA expression cassette nor its position in *HBB* IVSII affect transgene expression (Figure 2E). Similarly, the expression of vector-derived miR30 was also confirmed and found to be comparable across all cell populations transduced with shRNAmiR-encoding vectors (Figure 2F).

In addition, to ensure that the integration of the miR30-based shRNA-expression cassette within IVSII of the *HBB^AS3^* transgene did not lead to alternative mRNA splicing events, vector-derived *HBB^AS3^* mRNA levels were determined via RT-qPCR with primers designed to anneal specifically to exons 2 and 3 of this transcript (Appendix A). In cells transduced with the GLV2-βAS3, GLV2-βAS3-shSCRmiR and GLV2-βAS3-shGFPmiR vectors, a single RT-qPCR product of the expected size (132 bp) was observed, thus indicating that the insertion of the miR30-expression cassette at intron 2 maintains normal splicing of the *HBB^AS3^* transgene pre-mRNA (Appendix A).

Flow cytometry was employed for longitudinal analysis from the onset of erythroid differentiation changes in the eGFP levels in LV- and mock-transduced MEL-GFP cells (Figure 2G). This revealed an erythroid differentiation-dependent decrease in the GFP levels of GLV2-shGFPmiR- and GLV2-βAS3-shGFPmiR-transduced cells compared to the ubiquitous LV-Ef1a-shGFPmiR vector, where eGFP levels differed between LV-Ef1a-shGFPmiR- and LV-Ef1a-shSCRmiR-transduced cells from day 0 (Figure 2G). The percentage of eGFP-positive MEL-GFP cells, as measured on days 0, 7 and 10 of erythroid differentiation, was reduced by 27.0% (*p* < 0.01) and 35.9% (*p* < 0.0001) on days 7 and 10, respectively, for GLV2-shGFPmiR compared to GLV2-shSCRmiR. Likewise, the reduction in eGFP-positive MEL-GFP cells was 25.3% and 27.8% on days 7 and 10, respectively, for GLV2-βAS3-shGFPmiR compared to GLV2-βAS3-shSCRmiR. In contrast, the reduction of eGFP-positive cells was similar on days 0, 7 and 10 for the constitutively active Ef1a-promoter-driven LV-Ef1a-shGFPmiR LV compared to its SCR-equivalent, LV-Ef1a-shSCRmiR (37.7%, 33.1% and 48.4%, respectively, *p* < 0.0001, via one-way-ANOVA). For the parental vectors, GLV1-βAS3 and GLV2-βAS3, the percentage of eGFP-positive LV-transduced and mock-transduced MEL-GFP cells remained constant (87–99%) throughout differentiation.

The effects of the shRNA-encoding vectors on protein expression was also assessed for eGFP, HBB and endogenous Hba (as a marker of differentiation) by immunoblot analysis, using extracts from MEL-GFP cells on day 10 of erythroid differentiation (Figure 2H). Upon normalization for VCN/cell to adjust for variable gene transfer, a statistically significant decrease in GFP protein levels was observed across all sets of GFP-targeting vectors (53.79% decrease in eGFP for GLV2-shGFPmiR vs. GLV2-shSCRmiR, *p* < 0.001; 38.52% decrease in GFP for GLV2-βAS3-shGFPmiR vs. GLV2- βAS3-shSCRmiR, *p* < 0.01; and 49.75% decrease in GFP for LV-Ef1a-shGFPmiR vs. LV-Ef1a-shSCRmiR, *p* < 0.01, via one-way-ANOVA). Simultaneously, immunoblot analysis was also employed to determine vector-derived HBB^AS3^ protein levels (Figure 2H), based here and elsewhere in this manuscript on the co-detection of both HBB and HBB^AS3^ by anti-HBB antibody. Of note, equivalent vector-derived HBB^AS3^ levels were observed between the bifunctional (GLV2-βAS3-shSCRmiR or GLV2-βAS3-shGFPmiR)- and the parental vector (GLV2-βAS3)-transduced cells upon normalizing for VCN, thus leading us to conclude that the bifunctional vectors retain HBB^AS3^ expression levels similar to those of the GLV2-βAS3 vector.

In conclusion, these sets of data demonstrated that both the GLV2-shRNAmiR30 and the bifunctional GLV2-βAS3-shRNAmiR vectors produce functional shRNAmiRs, and confirmed the erythroid-specific expression of the shRNAmiR and *HBB^AS3^* transgene in bifunctional GLV2-βAS3-shRNAmiR vectors.

### 3.4. Validation of HBB^IVSI−110(G>A)^-Targeting Monofunctional GLV2-miR30shRNA and Bifunctional GLV2-βAS3-miR30shRNA LVs in MEL-HBB^IVS^ Cells

The design of RNApol(II)-driven RNAi vectors drew on the shRNA target sequence that had proven most effective for phenotypic correction based on U6-driven expression, that of shMID and shMID2 [22]. We thus employed shMID, plus two additional versions modified according to Guda and colleagues [44], by either shifting the first four bases of the guide sequence with GCGC nucleotides (shMIDA), or by replacing the first four bases of the passenger sequence with GCGC nucleotides (shMIDB), for insertion into the GLV2-shRNAmiR and the bifunctional GLV2-βAS3-shRNAmiR vector systems to generate GLV2-shMIDmiR, GLV2-shMIDAmiR, GLV2-shMIDBmiR, GLV2-βAS3-shMIDmiR, GLV2-βAS3-shMIDAmiR and GLV2-βAS3-shMIDBmiR, respectively (Figure 3A). Three additional vectors expressing shRNAmiR targeting the aberrant *HBB^IVSI−110(G>A)^* mRNA from the constitutively active pol(II)-based *Ef1a* promoter were also generated and labeled LV-Ef1a-shMIDmiR, LV-Ef1a-shMIDAmiR and LV-Ef1a-shMIDBmiR, respectively (Appendix A).

With a view to selecting appropriate candidates for further analysis in primary human CD34^+^ HSPCs, the humanized MEL-*HBB^IVS^* cell line, which had previously been demonstrated to faithfully represent *HBB^IVSI−110(G>A)^*-derived transcript expression ratios (40% aberrant mRNA compared with 46% in *HBB^IVSI−110(G>A)^*-homozygous CD34^+^), was used to assess the activity of GLV2-shRNAmiR and bifunctional GLV2-βAS3-shRNAmiR vectors in targeting aberrant *HBB^IVSI−110(G>A)^* mRNA and restoring normal HBB levels [22]. All five of the GLV2-shRNAmiR LVs, expressing shRNAmiR alone (i.e., GLV2-shMIDmiR, GLV2-shMIDAmiR, GLV2-shMIDBmiR, GLV2-shSCRmiR and GLV2-shGFPmiR) and the five bifunctional GLV2-βAS3-shRNAmiR LVs (i.e., GLV2-βAS3-shMIDmiR, GLV2-βAS3-shMIDAmiR, GLV2-βAS3-shMIDBmiR, GLV2-βAS3-shSCRmiR and GLV2-βAS3-shGFPmiR), concomitantly expressing shRNAmiR with the *HBB^AS3^* transgene, plus the parental GLV2-βAS3 and LV-U6-shMID/SCR LVs, were transduced individually into MEL-*HBB^IVS^* at MOI 3. The cells were then subjected to erythroid differentiation for 14 days. The short-hairpin-RNAmiR-mediated silencing activity of the LVs and the functional correction of the transduced cells were compared with mock-transduced cells at both mRNA and protein levels (Figure 3B).

Endogenous *HBB* and vector-derived miR30 and *HBB^AS3^* mRNA expression levels were assessed by RT-qPCR in LV- and mock-transduced MEL-*HBB^IVS^* cells collected on day 4 of differentiation (Figure 3C–E). The results obtained show no significant difference in Hba-normalized *HBB* mRNA levels when comparing mock- and LV-transduced cells for both the mono- and bi-functional vector sets, though considerable variation between experiments was noted (Figure 3C). Similarly, vector-derived *HBB^AS3^* mRNA was found to be retained between the parental GLV2-βAS3 and GLV2-βAS3-shRNAmiR-transduced samples after normalization for VCN/cell, although with substantial variation between experiments. Of note, the highest average *HBB^AS3^* mRNA expression, albeit without statistically significant differences, was observed for GLV2-βAS3-shMIDmiR-transduced cells (Figure 3D). Concomitantly, vector-derived miR30 expression, quantified across samples transduced with the GLV2-shRNAmiR or the bifunctional GLV2-βAS3-shRNAmiR vectors, revealed, on average, the equivalent expression of shRNAmiR from the two LV sets and confirmed, in combination with *HBB*-and β^AS3^-mRNA expression data, that the miR30-shRNA expression cassette does not interfere with vector-derived or endogenous *HBB*-like expression levels (Figure 3E). To investigate the knockdown efficacy of the shRNAmiRs, we then determined the transcript levels of normal to aberrant *HBB^IVSI−110(G>A)^* mRNA by RT-qPCR, with primers and probes that distinguish both transcript variants of *HBB*. Consistent with prior observations, no discernible difference of normal from aberrant *HBB* mRNA ratios was observed between GLV2-shRNAmiR and mock-transduced cells, and, in line with expectations, bifunctional GLV2-βAS3-shRNAmiR vectors significantly up-regulated normal to aberrant *HBB* mRNA ratios when compared to mock-transduced cells. However, bifunctional vectors only marginally increased the percentage of normal to aberrant *HBB* mRNA when compared with the parental GLV2-βAS3 vector (Figure 3F).

Immunoblot analysis for HBB and Hba from terminally differentiated mock- and LV-transduced MEL-*HBB^IVS^* cells was used to examine functional correction and the degree of differentiation of transduced cells, respectively (Figure 3G). Among the *HBB^IVSI−110(G>A)^*-specific GLV2-shRNAmiRs, only GLV2-shMIDmiR was found to be associated with slightly higher HBB levels, with the same pattern being observed for LV-U6-shMID-transduced cells. By contrast, comparative analysis of the three *HBB^IVSI−110(G>A)^* mRNA-specific shRNAmiRs expressed from the constitutively active *Ef1a*-promoter-driven LVs revealed that LV-Ef1a-shMIDmiR gave rise to a substantial induction of HBB in MEL-*HBB^IVS^* cells (Appendix A). Likewise, all GLV2-βAS3-shRNAmiR vectors were found to dramatically elevate HBB levels over mock-transduced cells, likely as a result of the expression of the *HBB^AS3^* transgene (Figure 3G). Across treatments, GLV2-AS3-shMIDmiR-transduced cells displayed the greatest increase in HBB protein levels (60% increase over mock control, *p* < 0.01) after normalization for VCN. Furthermore, significant HBB levels were also observed in other HBB *HBB^IVSI−110(G>A)^*-targeting bifunctional LVs. There was a lower-level induction of HBB detectable with GLV2-βAS3-shSCRmiR, possibly due to stress-induced changes in globin expression (Figure 3H). The degree of erythroid differentiation, as determined by the immunoblot detection of Hba, was comparable between samples (Figure 3I).

Based on these sets of results, the bifunctional GLV2-βAS3-shMIDmiR and its, albeit low-performing, RNAi-only equivalent GLV2-shMIDmiR LVs were selected for the further evaluation of their therapeutic activity compared to the parental GLV2-βAS3, in primary patient-derived CD34^+^ HSPCs, as clinically relevant substrates.

### 3.5. Efficacy of LVs Expressing miR30^shRNA^ under the HBB Promoter Alone or as Part of IVSII of the HBB^AS3^ Transgene in HBB^IVSI−110(G>A)^-Homozygous Primary CD34^+^ Cells

The therapeutic potential of GLV2-shMIDmiR and GLV2-βAS3-shMIDmiR LVs (and respective controls) and the parental GLV2-βAS3 LV was further assessed in vitro by the transduction of HSPCs derived from the peripheral blood of three *HBB^IVSI−110(G>A)^*-homozygous β-thalassemia individuals (Figure 4A). Specifically, two patient-derived HSPC samples (P1, P2) were transduced at a therapeutically relevant MOI of 3 and assessed for transgene expression and correction of the β-thalassemia phenotype. In parallel, HSPCs obtained from a third *HBB^IVSI−110(G>A)^*-homozygous patient (P3) were transduced at high MOI (MOI > 10) to assess potential changes in the outcome when shRNAmiRs are highly expressed. Starting 48 h post-transduction, the cells were differentiated for 9 days, during which time they were counted to determine cell expansion. Cytocentrifugation of the samples was undertaken to assess the degree of differentiation after May–Grünwald–Giemsa and dianisidine staining (Figure 4B). In addition, flow cytometry analysis was performed on the first and last days of differentiation to determine potential changes in apoptosis and cell death (based on combined staining with propidium iodide and YO-PRO-1). No discernible differences in the viability of the cells transduced at either low or high LV MOI was observed (Figure 4C).

HPLC analysis was employed to determine the relative production of HbA and/or HbA^βAS3^ levels in the terminally differentiated erythroid cells of P1 and P2 samples. The results showed no appreciable differences in HbA levels between the GLV2-shMIDmiR- and mock-transduced cells, while GLV2-βAS3-shMIDmiR was found to produce amounts of HbA|HbA^βAS3^ that were comparable to the parental GLV2-βAS3 vector. Interestingly, in terminally differentiated erythroid cells of the P3 sample (at average VCN > 10 per cell), GLV2-shMIDmiR was found to produce slightly higher HbA levels than the mock-transduced control, while GLV2-βAS3-shMIDmiR gave rise to a similar induction of HbA|HbA^βAS3^ as the parental GLV2-βAS3. This suggests the need for a high VCN/cell for RNApol(II)-driven shRNAmiRs to mediate an effect against *HBB^IVSI−110(G>A)^* transcripts. Furthermore, equivalent proportions of HbA|HbA^βAS3^ levels were observed between patient P1/P2- and P3-derived HSPCs transduced with the parental GLV2-βAS3 at low and high MOI, respectively, which likely indicates a saturation effect of the parental GLV2-βAS3 vector on HbA|HbA^βAS3^ induction, irrespective of the VCN/cell above a threshold or the presence of shRNAmiR-mediated *HBB^IVSI−110(G>A)^* transcript silencing (Figure 4D).

Further quantification of the vector-derived *HBB^AS3^* transgene mRNA expression at low MOI in P1 and P2 HPSC samples and normalization of the results for VCN/cell revealed a markedly lower expression in cells transduced with the bifunctional GLV2-βAS3-shMIDmiR and GLV2-βAS3-shSCRmiR LVs compared to cells transduced with the parental GLV2-βAS3, while the endogenous *HBB* mRNA expression remained unaffected between samples (Figure 4E–G). In line with the notion of the saturating effects of GLV2-βAS3-derived expression, in the P3 patient sample (transduced at high MOI), an equivalent expression of the *HBB^AS3^* transgene was observed between cells transduced with the bifunctional and the parental LVs. Interestingly, further measurement of vector-derived miR30 mRNA between the monofunctional GLV2-shRNAmiR and bifunctional GLV2-βAS3-shRNAmiR LVs also revealed a somewhat lower expression of miR30 in mono- over bifunctional LVs at high MOI (Figure 4H). Importantly, the RT-qPCR-based absolute quantification of the total transcript levels of normal to aberrant *HBB^IVSI−110(G>A)^* mRNA revealed that the parental GLV2-βAS3 vector consistently induced a higher normal-to-aberrant *HBB* mRNA ratio than other LVs (Figure 4I).

Taken together and in line with findings in MEL cells, the results obtained in patient-derived HSPCs further emphasize the superior gene expression efficiency of the GLV2-βAS3 LV over the monofunctional shRNAmiR-expressing and bifunctional βAS3-globin/shRNAmiR-expressing vectors.

## 4. Discussion

Gene therapy clinical trials for β-thalassemia, focusing on compensating for the inherited mutated *HBB* through the addition of a normal functioning copy, have been shown to improve the disease phenotype effectively. However, it is also evident from clinical trials that HbA synthesis at therapeutically relevant levels in individuals harboring *HBB^IVSI−110(G>A)^* and in the majority of β^0^/β^0^-thalassemic patients remains an unmet clinical need. Despite significant progress, the low LV transduction efficiency of HSPCs frequently prevents a clinically meaningful expression of the *HBB* transgene from being achieved [5,7,8,9,10]. Thus, the generation of LVs that combine high transfer efficacy with clinically relevant expression levels of the *HBB* transgene at low VCNs has long been a key developmental goal.

Thus far, major breakthroughs in LV designs that enabled more efficient HSPC transduction and the expression of *HBB* transgenes in HSPC-derived red blood cells include (a) the identification and removal of DNA elements that negatively affected the viral titer [45], (b) the inclusion of enhancer-blocking or barrier insulator elements from the chicken βLCR (CHS4) [46,47,48,49] and human ankyrin 1 gene promoter [50,51], which protect the transgene from potentially negative effects from the endogenous repressive genetic and epigenetic elements at the sites of LV integration and (c) the inclusion of the human βLCR element HS1 in addition to the HS2, HS3 and HS4 sites to promote higher expression levels [52,53]. Additionally, the replacement of the normal human *HBB* gene with a mutant gene bearing one (*HBB^T87Q^*) or three (*HBB^G16D,E22A,T87Q^*) anti-sickling amino acid substitutions, which confer greater affinity for α-globin (G16D) or reduce interactions favoring protein precipitation (E22A, T87Q), has been shown to further improve the sickle cell phenotype [35]. Nevertheless, ongoing LV-based clinical studies for β-hemoglobinopathies underline the requirement for a high VCN/cell for therapeutically relevant transgene expression to be achieved in patients with severe β-thalassemia (e.g., β^0^/β^0^ genotype) and sickle cell disease, where abundant sickling β^S^-globin competes with vector-derived β-like globin for hemoglobin incorporation [4,9]. However, elevated VCNs in HSPCs increase the risk of insertional mutagenesis and thus of β-hemoglobinopathy patients developing treatment-related hematological malignancies.

In the present study, we report the development of the GLV2-βAS3 LV, which, compared to the original GLOBE vector currently undergoing evaluation in clinical trials [5] and the intermediate GLV1-βAS3 LV, downstream of *HBB* bears an 848-bp sequence containing the *HBB* transcription terminator region for more efficient pre-mRNA 3′ end formation [27,28] and thus the greater accumulation of mature *HBB* cytoplasmic mRNA. A major advantage of the GLV2-βAS3 vector is that, in addition to retaining high vector titer production, it allows for superior β-like globin expression in primary patient-derived HSPCs, including those with a *HBB^IVSI−110(G>A)^* β-thalassemia genotype. Although not completely suppressed, the reduction in aberrant *HBB^IVSI−110(G>A)^* mRNA observed in homozygous *HBB^IVSI−110(G>A)^* patient-derived HSPCs treated with the GLV2-βAS3 vector could be beneficial for many β-hemoglobinopathy-causing mutations affecting RNA expression and processing. With its abundance and stable expression of vector-derived HBB^AS3^, the GLV2-βAS vector may be highly suitable for a range of β-hemoglobinopathies, including those with the complete absence or aberrant processing of HBB transcripts. Additional side-by-side comparison studies using HSPCs from sickle cell or β^0^/β^0^ β-thalassemia donors would allow for the corresponding assessment of the correction of red blood cell morphology and hematologic parameters by the GLV2-βAS3 LV in these more therapeutically challenging populations.

Furthermore, the development of dual-function LVs combining *HBB*-like transgene expression with potent shRNAmiRs designed in independent studies to selectively target the mRNA of (a) *BCL11A* to combine de-repression of *HBG* with expression of *HBB* [54,55], (b) *HBA2* to emulate an α-thalassemia carrier background [56] and (c) *HBB^S^ (*β^S^) to reduce β^S^-encoding mRNA *(HBB Glu6Val)* in sickle cell disease patients [23,57], has paved the way for more targeted, mutation-specific gene therapy approaches to β-thalassemia. Earlier findings by our group based on the RNAi of aberrant *HBB^IVSI−110(G>A)^* mRNA in HSPCs from *HBB^IVSI−110(G>A)^* β-thalassemia individuals firmly established the aberrant transcript as a rational target for a mutation-specific gene therapy approach [22]. We therefore hypothesized that the development of a gene therapy strategy that combines the silencing of the aberrantly spliced *HBB^IVSI−110(G>A)^* mRNA with the *HBB^AS3^* transgene would be of significant therapeutic relevance for *HBB^IVSI−110(G>A)^* β-thalassemia patients in the Mediterranean region, in which this genotype is very common.

The analyses undertaken here were restricted to those that would add further information to the vectors with well-characterized behavior and as appropriate for the cell substrates employed. The backbones of the GLOBE [34] and pLKO.1-TRC [58] vectors and their expression cassettes have been widely analyzed in human cells, including long-term repopulating HSCs [59] and embryonic stem cells [22], respectively, and in preparation for clinical trials on the GLOBE vector [5], they have also been analyzed in clonogenic assays for the retrospective evaluation of their oligopotency and proliferation potential [59]. Investigation of the potential changes in vector properties, such as integration patterns and their influence on the differentiation and proliferation potential, in connection with the alternative inserts applied here, will be of critical importance before any clinical application, but would not be faithfully represented by the MEL cells or the expanded CD34^+^ cells employed in the present study. Such analyses, and additional long-term repopulation and lineage analyses after transplantation in immunodeficient mice, is restricted to studies based on mobilized or bone-marrow aspirated cell samples. Instead, we focused here on transduction efficiencies and the assessment of phenotype correction in the erythroid lineage at the mRNA, protein and cell-morphology level.

LVs expressing the miRNA-30-based shRNA expression cassette from a βLCR *HBB* promoter combination were used to evaluate different shRNAmiR designs individually and assess the efficacy of the shRNAmiR-mediated silencing of the aberrant *HBB^IVSI−110(G>A)^* mRNA as a monotherapy for *HBB^IVSI−110(G>A)^* β-thalassemia. Similarly, the simultaneous evaluation of the bifunctional GLV2-βAS3-shRNAmiR set of vectors combining *HBB^AS3^* transgene expression with the IVSII-encoded shRNAmiR-mediated silencing of the aberrant *HBB^IVSI−110(G>A)^* mRNA, was performed to assess potential superiority of the bifunctional over the parental GLV2-βAS3 LV for vector-derived HbA^βAS3^ production and for the decreased expression of aberrant endogenous *HBB* mRNA products. Our results demonstrated that vector titers and the transduction efficiency of the monofunctional GLV2-shRNAmiR and of bifunctional GLV2-βAS3-shRNAmiR were equivalent to those of the parental GLV2-βAS3 gene therapy vector in MEL-*HBB^IVS^* cells and primary human hematopoietic CD34^+^ cells. This confirms that the inclusion of the miR30 expression cassette had no effect on the production and functionality of the GLV2-βAS3 LV.

Among the different *HBB^IVSI−110(G>A)^*-specific shRNAmiRs tested in MEL-*HBB^IVS^*, shMIDmiR was found to increase HBB protein levels only marginally, whereas modifications to the original *HBB^IVSI−110(G>A)^*-specific shRNA sequence based on previous findings [44] had no additional effect, in agreement with the observations of Brusson and colleagues [23]. Additionally, in line with our previous report [22], *HBB^IVSI−110(G>A)^*-specific shRNAmiRs in MEL-*HBB^IVS^* did not correct aberrant-to-normal mRNA ratios, likely indicating a post-translational inhibition rather than a degradation of the aberrant mRNA being responsible for the increased HBB protein levels in GLV2-shMIDmiR-transduced cells. The late erythroid expression of shRNAmiR from the *HBB* promoter may be a key cause for the lower normal HBB production following monofunctional GLV2-shRNAmiR vector administration compared to LVs that instead drive high-level constitutive expression from the U6 or *Ef1a* promoters.

Immunoblot and RT-qPCR analyses of the GLV2-βAS3-shRNAmiR-transduced MEL-*HBB^IVS^* cells revealed that all GLV2-βAS3-shRNAmiR vectors achieve a reduction in aberrant *HBB^IVSI−110(G>A)^* mRNA with a concomitant, significantly higher production of HBB chains compared to the mock-transduced MEL cells, indicating that HBB^AS3^ outcompetes endogenous HBB protein production. The observation that, at similar VCNs, the bifunctional GLV2-βAS3-shRNAmiR LVs achieve similar phenotypic correction and HBB|β^AS3^ expression compared to the parental GLV2-βAS3 vector, indicated that the addition of shRNAmiR did not add to the therapeutic efficiency of the vectors. Whether this is attributable to either (a) the late erythroid expression of shRNAmiR, (b) inefficient pre-mRNA processing compared to U6-promoter-driven shRNAs of IVSII-encoded *HBB^IVSI−110(G>A)^*-specific shRNAmiR, or (c) a combination of these effects, remains to be investigated. Similarly, the inefficient silencing of aberrant *HBB* mRNA for our RNApol(II)-driven shRNAmiRs compared to the efficient knockdown of *BCL11A* mRNA as previously reported for similar vectors that combine a *HBB^AS3^* transgene addition approach with *BCL11A*-targeting shRNAmiR-based silencing [54,55], may in part be explained by the high abundance of aberrant *HBB^IVSI−110(G>A)^* mRNA in comparison to the much lower mRNA expression levels of the BCL11A transcription factor. While other intronic sites for insertion of the shRNAmiR were not tested in the present study, which could in principle also include *HBB* IVSI [54], the vector-derived mRNA expression for the bifunctional vectors was comparable to, and at times exceeded, that of the GLV2-βAS3 LV, demonstrating a neutral effect of the chosen shRNAmiR *HBB* IVSII insertion site on transgene expression. The lack of a therapeutic benefit of the GLV2-βAS3-shMIDmiR LV compared to the parental vector GLV2-βAS3 may be attributable to the latter’s strong performance. Accordingly, the timing and level of expression, in conjunction with the efficient pre-mRNA processing resulting from the inclusion of the *HBB* transcriptional terminator region, may result in the production of sufficiently high mRNA expression and the translation of functional HBB chains [60] before levels of aberrant mRNA reach a point where translation factors become limiting, resulting in inefficient *HBB* mRNA translation. Finally, the comparable HbA|HbA^βAS3^ levels observed between patient-derived HSPCs transduced with the GLV2-βAS3 at low or high MOI suggests a saturation of vector-derived expression at low MOI and VCNs, thus highlighting the exceptional potential of the GLV2-βAS3 LV for the treatment of severe β-hemoglobinopathies.

## 5. Conclusions

In an effort to enhance the therapeutic potential of GLOBE-based LV gene addition therapies without relying on elevated VCNs and thus genotoxic risks, this study exploited the use of the *HBB* transcription terminator element inserted downstream the *HBB^AS3^* transgene. This resulted in higher cytoplasmic mRNA (4- to 9-fold in MEL cells and 2-fold in human CD34^+^ HPSCs upon induced erythroid differentiation) and protein (3- to 4-fold in MEL cells and 2-fold in CD34^+^ HPSCs) levels. The GLV2-βAS3, incorporating the 809-bp β-Term sequence, thus substantially outperforms the original GLOBE LV and vector designs mediating the RNAi of aberrant *HBB^IVSI−110(G>A)^* mRNA. GLV2-βAS3 retains high vector titers in vector production and allows for superior *HBB*-like expression and the functional correction of the β-thalassemia phenotype in primary patient-derived HSPCs.

## Figures and Tables

**Figure 1 cells-12-02848-f001:**
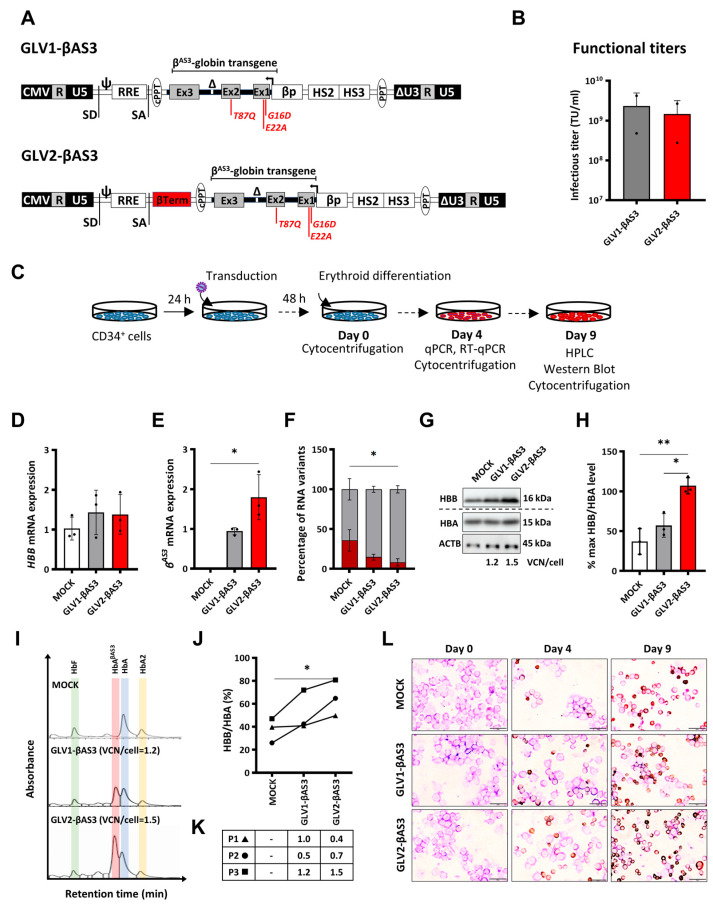
Design and functional characterization of the GLOBE-based *HBB^AS3^* transgene-expressing lentiviral vectors in HSPCs. (**A**) Schematic representation of the novel GLV1-βAS3 and GLV2-βAS3 vectors, the latter incorporating the *HBB* transcription termination sequence. CMV, cytomegalovirus immediate early promoter; R, repeated region found within both 5′ and 3′ LV long terminal repeats (LTRs); U5, unique 5′ region; SD, splicing donor site; ψ, packaging sequence; RRE, HIV-1 Rev-responsive element; SA, splicing acceptor site; cPPT, central polypurine tract; Ex1/2/3, human *HBB* exons; βTerm, 848-bp *HBB* transcription terminator sequence downstream of *HBB*; Δ, 593-bp deletion within *HBB* IVSII; βp, *HBB* promoter; βLCR HS2 and HS3, DNase I-hypersensitive sites; PPT, polypurine tract; ΔU3, 3′ self-inactivating LTR. The three anti-sickling amino-acid substitutions (G16D, E22A and T87Q) in the β^AS3^ transgene are indicated in red. (**B**) The bar chart shows the infectious titers of GLV1-βAS3 and GLV2-βAS3 LVs expressed as transducing units per mL (TU/mL) based on two independent viral preparations. Data are presented as mean ± SD. (**C**) *HBB^IVSI−110(G>A)^*-homozygous β-thalassemia patient-derived HSPC transduction with GLV1-βAS3 and GLV2-βAS3 at low MOI and erythroid differentiated for 9 days to assess vector-derived β^AS3^-transgene expression potential. (**D**) Relative endogenous *HBB* mRNA expression and (**E**) vector-derived *HBB^AS3^* mRNA expression (n = 3) measured by RT-qPCR on day 4 of erythroid differentiation. Endogenous and vector-derived *HBB^AS3^* mRNA levels normalized against *HBA* (combined *HBA1* and *HBA2*) mRNA are further normalized for VCN. (**F**) Percentage contribution of aberrant and normal *HBB* mRNA in bulk-transduced HSPCs (n = 3). (**G**) Representative images of immunoblots of HBB and HBA (HBA1 and HBA2) proteins and ACTB (β-actin, used as loading control) in protein extracts of terminally differentiated *HBB^IVSI−110(G>A)^* patient-derived HSPCs. The dashed line indicates detection on separate membranes. (**H**) Percentage of differentiation-normalized HBB levels (normalized for VCN) following densitometry analysis of immunoblots from three independent transduction experiments on *HBB^IVSI−110(G>A)^*-homozygous patient-derived HSPCs. Statistical significance was calculated using Mann–Whitney-U test, where * indicates *p* < 0.05 and ** *p* < 0.01. (**I**) Representative histograms of hemoglobin variants in the soluble cellular fraction of terminally differentiated mock- and LV-transduced erythroid cells as measured by cation-exchange HPLC. (**J**) Percentage of differentiation-normalized HBB globin chain (normalized for VCN) as detected by reversed-phase HPLC analysis of the soluble cellular fraction obtained from terminally differentiated erythroid cells. Statistical significance was calculated using one-way ANOVA, * *p* < 0.05 (n = 3). (**K**) VCN measurements for the three independent replicates, with symbols indicating the corresponding HBB/HBA curve identities from panel 1(**J**). (**L**) Representative images of May–Grünwald–Giemsa- and dianisidine-stained cytocentrifugation samples of mock- and LV-transduced *HBB^IVSI−110(G>A)^*-homozygous CD34^+^ cells on days 0, 4 and 9 of erythroid differentiation. Magnification: 800×; scale bars, 20 μm.

**Figure 2 cells-12-02848-f002:**
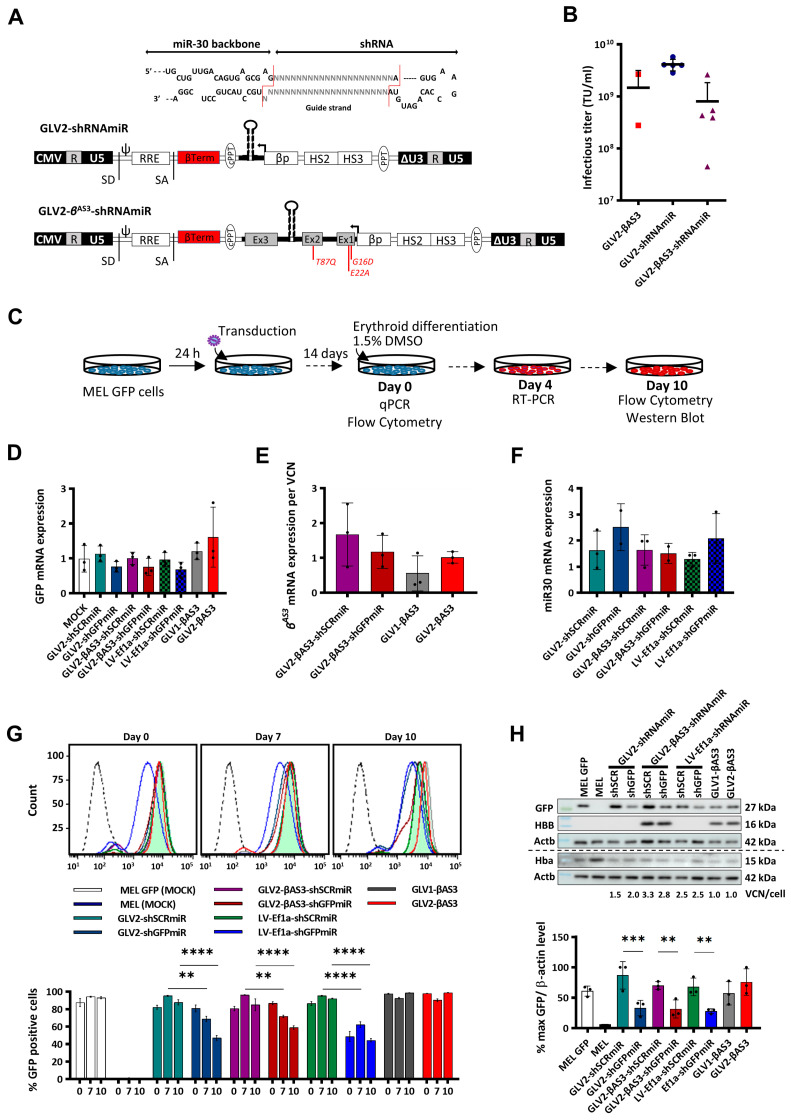
Generation and functional validation of GLOBE-based lentiviral vectors expressing a miR30-based shRNAmiR expression cassette alone (GLV2-shRNAmiR) or in combination with the *HBB^AS3^* transgene (bifunctional GLV2-βAS3-shRNAmiR). (**A**) Schematic representation of GLV2-shRNAmiR structure in its proviral form, modified to express the shRNAmiR expression cassette downstream of the *HBB* control elements in place of the *HBB^AS3^* transgene to achieve erythroid-specific RNApol(II)-driven miR30shRNA production, and into IVSII of the *HBB^AS3^* transgene to produce the βAS3/shRNAmiR bifunctional vector (GLV2-βAS3-shRNAmiR). Magnified above, shRNAs consisting of two 21-bp stems linked by a 19-base loop were inserted into an miR30 scaffold flanked by a 125-bp miR30 flanking region on either side of the hairpin. (**B**) Vector titers of harvested lentiviral vectors expressed as transducing units per mL (TU/mL) following 350-fold concentration and calculated via multiplex qPCR (single production per vector). (**C**) Overview of MEL-GFP cell transduction with GLV2-shRNAmiR and GLV2-βAS3 at increasing MOI (0.5, 1, 2) and erythroid differentiation for 10 days to assess vector-derived *HBB^AS3^* transgene expression levels. (**D**) Relative GFP, (**E**) vector-derived *HBB^AS3^* and (**F**) vector-derived miR30 mRNA expression measured via RT-qPCR on day 4 of erythroid differentiation. Target mRNA levels were normalized against *Hba* mRNA and expressed per vector copy (n = 3). (**G**) (top) Representative histograms of GFP expression and (bottom) comparison of the fluorescence values (n = 3) of MOCK- and LV-transduced MEL-GFP cells at days 0, 7 and 10 of erythroid differentiation. (**H**) Representative images of immunoblots for the measurement of GFP, HBB and Hba against Actb as loading control in transduced MEL-GFP cells on day 10 of erythroid differentiation, based on equivalent detection of HBB and HBB^AS3^ by the anti-HBB antibody. Below, corresponding relative quantities of GFP (normalized to Actb) based on densitometry of immunoblots across multiple experiments (ImageJ v1.53p). The dashed line indicates detection on separate membranes. Data represent mean ± SD of three independent experiments from a single LV preparation. Statistical significance was calculated using one-way ANOVA; **, *p* < 0.01; ***, *p* < 0.001; and ****, *p* <0.0001.

**Figure 3 cells-12-02848-f003:**
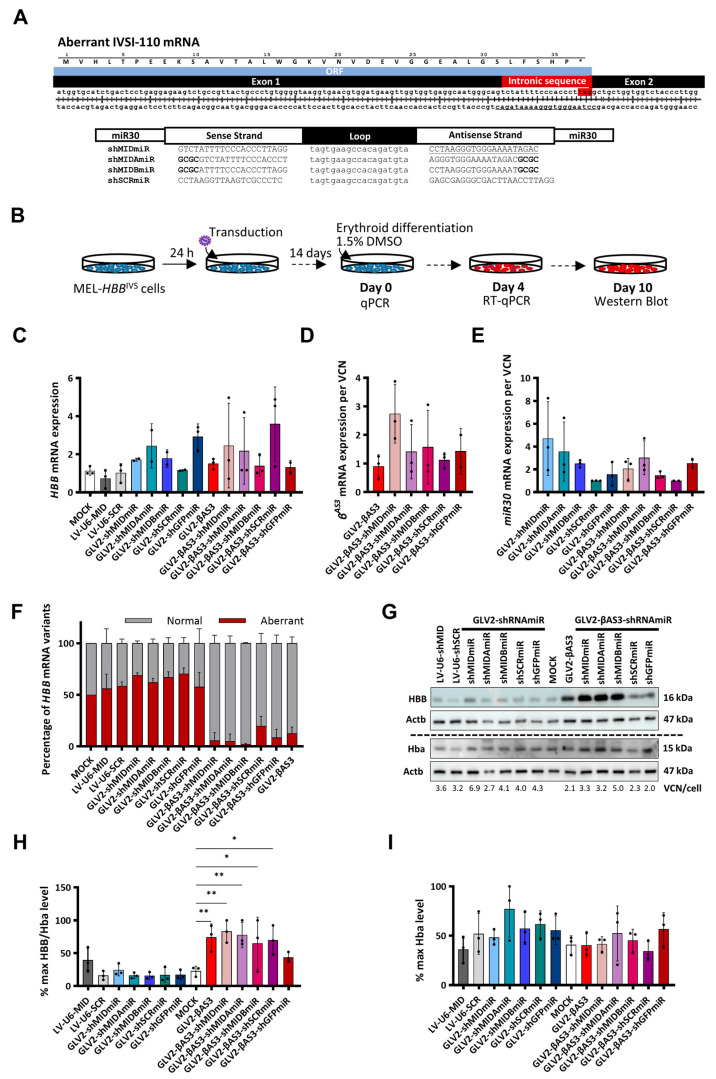
Design and functional evaluation of the shRNAmiR guide strand variants targeting the aberrant *HBB^IVSI−110(G>A)^* mRNA using the GLV2-shRNAmiR and bifunctional GLV2-βAS3-shRNAmiR vectors. (**A**) (top) Schematic of double-stranded cDNA of the aberrant *HBB^IVSI−110(G>A)^* transcript, with a stop codon (*) in the retained 19-nt intronic sequence concluding the open reading frame (ORF). (bottom) shRNAs targeting the aberrant *HBB^IVSI−110(G>A)^* mRNA consisting of two 21-bp stems linked by a 19-base loop were inserted into an miR30 scaffold flanked by a 125-bp miR30 flanking region on either side of the hairpin. (**B**) Overview of MEL-*HBB^IVS^* cell transduction with GLV2-shRNAmiR and GLV2-βAS3-shRNAmi at MOI = 3 and erythroid differentiation for 10 days to assess target gene knockdown efficiency and vector-derived *HBB^AS3^* transgene expression levels. (**C**) Relative expression of human *HBB* mRNA (**D**), vector-derived *HBB^AS3^* mRNA and (**E**) vector-derived miR30 mRNA expression (n = 3) measured by RT-qPCR on day 4 of erythroid differentiation. Target mRNA levels were normalized against endogenous *Hba* mRNA and expressed per LV copy number. (**F**) Percentage contribution of aberrant and normal *HBB* mRNA in bulk MEL-*HBB^IVS^* cells (n = 3). (**G**) Representative images of immunoblots for HBB, Hba and Actb (loading control) protein levels in transduced MEL-*HBB^IVS^* cells on day 10 of erythroid differentiation, based on the equivalent detection of HBB and HBB^AS3^ by anti-HBB antibody. The dashed line indicates detection on separate membranes. (**H**) Percentage of differentiation-normalized HBB levels (also normalized for VCN) and (**I**) Actb-normalized Hba levels relative to the highest value for each experiment as determined by densitometry analysis of the immunoblots from three independent transduction experiments on MEL-*HBB^IVS^* cells. Statistical significance was calculated using Mann–Whitney-U test, where * indicates *p* < 0.05 and ** *p* < 0.01.

**Figure 4 cells-12-02848-f004:**
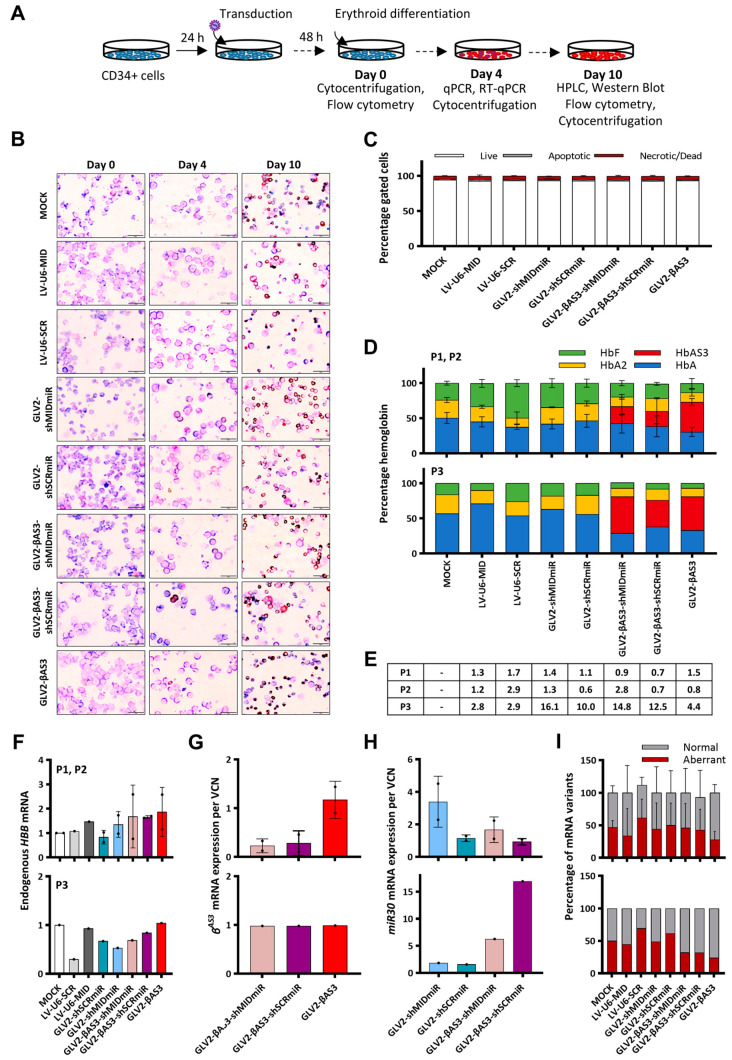
Evaluation of shRNAmiR-expressing vectors in primary human erythroid cells. (**A**) Overview of the transduction of *HBB*^IVSI−110(G>A)^-homozygous β-thalassemia patient-derived HSPCs with GLV2-shRNAmiR or bifunctional GLV2-βAS3-shRNAmiR at low (3) and high (10) MOI and erythroid differentiation for 10 days to assess vector-derived *HBB^AS3^* transgene expression and correction of the β-thalassemia phenotype. (**B**) Representative images of May–Grünwald–Giemsa- and dianisidine-stained cytocentrifugation samples of mock- and LV-transduced HSPCs on days 0, 4 and 10 of erythroid differentiation. (**C**) Percentage of live, apoptotic and dead cells, as determined by flow cytometry analysis following combined staining with propidium iodide and YO-PRO-1. Data are presented as mean ± SD of three independent experiments. (**D**) Stacked bar charts show percentage of hemoglobin variants as detected by HPLC analysis of the soluble cellular fraction obtained from terminally differentiated erythroid cells transduced at low (P1, P2) and high (P3) MOI. (**E**) VCN measurements for the three independent replicates. (**F**) Relative endogenous *HBB* mRNA expression and (**G**) vector-derived *HBB^AS3^* and (**H**) miR30 mRNA expression measured by RT-qPCR on day 4 of erythroid differentiation. Endogenous and vector-derived *HBB^AS3^* mRNA levels normalized against *HBA* mRNA were expressed per vector copy. (**I**) Percentage contribution of aberrant and normal *HBB* mRNA in bulk-transduced HSPCs (n = 3).

## Data Availability

The data that support the findings of this study are available upon request.

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
