# Peer review of "Evaluation of Mono- and Bi-Functional GLOBE-Based Vectors for Therapy of β-Thalassemia by HBBAS3 Gene Addition and Mutation-Specific RNA Interference"

_cells, 2023, doi:10.3390/cells12242848_

Round 1

Reviewer 1 Report

Comments and Suggestions for Authors

Koniali et al show a comprehensive assessment of lentiviral vectors developed for treatment of beta-thalassemia. Multiple vector elements were tested including a terminator sequence in combination with microRNA shRNA sequences to specifically improve efficacy in HBBIVSI−110(G>A) red blood cells leading to single bifunctional vector showing promising results. The final LV vector resulted in high titers with therapeutic utility of late-erythroid HBBIVSI−110(G>A)-specific miR30-shRNA expression, and highlighted the potential of GLV2-βAS3 for the treatment of severe β-hemoglobinopathies.

Minor comments:

-       Line 238 – 241 is a different font type.

-       In Figure S3A ‘WRPE’ should be ‘WPRE’

-       The (Patsali et al. 2018) reference may be mentioned in the Introduction section, not the abstract (line 32).

-       Alexis A. Thompson is referenced multiple times, but change to ‘Thompson et al’, consistent with the other references (just using surnames).

-       In the references list, the references are inconsistently stated, sometimes with initials and sometimes with full names. Please make it consistent according to journal policy.

Author Response

Koniali et al show a comprehensive assessment of lentiviral vectors developed for treatment of beta-thalassemia. Multiple vector elements were tested including a terminator sequence in combination with microRNA shRNA sequences to specifically improve efficacy in HBBIVSI−110(G>A) red blood cells leading to single bifunctional vector showing promising results. The final LV vector resulted in high titers with therapeutic utility of late-erythroid HBBIVSI−110(G>A)-specific miR30-shRNA expression, and highlighted the potential of GLV2-βAS3 for the treatment of severe β-hemoglobinopathies.

[Reply] We agree with the Reviewer’s summary and are grateful for the overall positive assessment.

Minor comments:

-    Line 238 – 241 is a different font type.

[Reply] Corrected.

-    In Figure S3A ‘WRPE’ should be ‘WPRE’

[Reply] Done.

-    The (Patsali et al. 2018) reference may be mentioned in the Introduction section, not the abstract (line 32).

[Reply] The literature reference has been removed from the abstract.

-    Alexis A. Thompson is referenced multiple times, but change to ‘Thompson et al’, consistent with the other references (just using surnames).

[Reply] With apologies for the oversight during first submission, the reference style has now been changed to indices in square brackets, as suggested by the MDPI Style Guide and in line with the style used by the American Chemical Society. This change in format has simultaneously corrected inconsistencies for citations in our original submission, including our mistaken citation here of the author’s first name.

-    In the references list, the references are inconsistently stated, sometimes with initials and sometimes with full names. Please make it consistent according to journal policy.

[Reply] Thank you for the vigilance; the inconsistencies have been addressed by an overall change and revision of our citatio format; see the previous comment.

Reviewer 2 Report

Comments and Suggestions for Authors

In this study, the authors use gene therapy by addition of the anti-sickling βAS3-globin transgene as curative method for patients with β-thalassemia. They use different strategies to improve the transduction and transcription of the βAS3-globin to enhance therapeutic potential without increasing genotoxic risks. The results and synthesis of the constructs are well described However, the manuscript need improvement.

1-     Remove the reference cited in the Abstract

2-     The authors need to give a short definition about β-hemoglobinopathies and thalassemia in the introduction

3-     There are different MEL cell line, please define which one exactly (SK-MEL-28, SK-MEL-31, SK-MEL-1….etc). For the citation of cell line please follow the ATCC manner.

4-     For the origin off all reagents, describe the company, the city and country (like SFEM II medium (Stem cell Technologies, Vancouver, Canada)). This should be uniform in whole manuscript

5-     The line 237-242, the text has different format

6-     Since the development of leukemia in patient after LV therapy (Science. 2003; 302: 415-419, J Clin Invest. 2008; 118: 3143-3150) the interesse in LV is significantly reduced. Can you explain why you don’t use AAV or ADV instate of LV System?

7-     In the results part, you should describe your results, no reference should be cited. Explanation and comparison with other works must be in the discussion part. In the result part 3.1 it is enough if you start with “To evaluate whether the extra….”. From line 269-281 you can delete it.

8-      In part 3.2, no need to describe previous work, this you can make it in the discussion. You can start the part 3.2 with “Hypothesizing that silencing of the aberrantly….”

9-     For part 3.2.2, the same like previously, no need to describe previous work.

1-  The data availability statement should be changed, the authors generate many data, the row data should be available for reader on reasonable request.

1-  The authors don’t analyze the constitution of the blood cells, it will be of great interest, if they show that the transduction doesn’t influencing the constitution of blood cells, at less analyze the constitution of the lymphocyte subset. Could you explain, why you don’t do that?

Comments on the Quality of English Language

The english quality is fine, it needs only minor editing

Author Response

In this study, the authors use gene therapy by addition of the anti-sickling βAS3-globin transgene as curative method for patients with β-thalassemia. They use different strategies to improve the transduction and transcription of the βAS3-globin to enhance therapeutic potential without increasing genotoxic risks. The results and synthesis of the constructs are well described However, the manuscript need improvement.

[Reply] We thank the Reviewer for the overall positive assessment and for corrections and suggestions for this manuscript.

1-       Remove the reference cited in the Abstract

[Reply] Done.

2-       The authors need to give a short definition about β-hemoglobinopathies and thalassemia in the introduction

[Reply] The Introduction now begins with a sentence defining β-thalassemia.

3-       There are different MEL cell line, please define which one exactly (SK-MEL-28, SK-MEL-31, SK-MEL-1….etc). For the citation of cell line please follow the ATCC manner.

[Reply] Done. In section 2.2, we have added more specifically “APRT(-) cell line C88”, in line with how the cells were first described in Antoniou 1991 Methods Mol Biol (PMID: 21416373). Unable to find an exact equivalent with ATCC and if the additional self-citation is allowed, we will be happy to add the corresponding reference in the manuscript. 

4-       For the origin off all reagents, describe the company, the city and country (like SFEM II medium (Stem cell Technologies, Vancouver, Canada)). This should be uniform in whole manuscript

[Reply] Done.

5-     The line 237-242, the text has different format

[Reply] This has been corrected.

6-       Since the development of leukemia in patient after LV therapy (Science. 2003; 302: 415-419, J Clin Invest. 2008; 118: 3143-3150) the interesse in LV is significantly reduced. Can you explain why you don’t use AAV or ADV instate of LV System?

[Reply] The publications indicated by the Reviewer and the tendency of the γ-retroviruses used in the corresponding clinical trials to integrate close to transcriptional start sites of endogenes and thus to readily activate proto-oncogenes, such as LMO2, greatly troubled the gene therapy field at the time. Gene addition therefore switched to safer lentiviral vectors, as used in our study, which have a more random integration profile, albeit still with a preference for open chromatin structures (Cattoglio et al 2007 Blood (PMID: 17507662)), with the added benefit that in contrast to γ-retroviruses, lentiviruses also effectively transduce non-dividing cells, which greatly enhances their suitability for the modification of often quiescent, true hematopoietic stem cells.

By contrast, adenoviral vectors in the vast majority of cases remain episomal and though effective vectors for other applications are therefore not suitable for curative treatment of highly proliferative HSPCs. Similarly, adeno-associated viral vectors are ineffective vehicles for permanent integration in their own right, although their ssDNA genome has lately made them effective vectors for donor templates in combination with genome editing. For the application of HBB|HBBAS3 gene addition, however, their payload of below 4.5 kb is a severe limitation, whereas mitigation of this shortcoming by using dual-vector approaches lowers efficiency.

For ex vivo gene addition in the hematopoietic system, lentiviral vectors are therefore currently the preferred option in clinical trials and most research applications. Though lentiviral vectors have thus far correspondingly proven exceptionally safe and efficient for application to hemoglobinopathies, we should point out for full disclosure that a bluebird bio phase 3 trial to treat cerebral adrenoleukodystrophy was linked to the development of myelodysplastic syndrome in trial subjects in 2021. It appeared, however, that use of a constitutive promoter (in contrast to the erythroid promoter used in our present study) was a key contributor to adverse events by inadvertent activation of nearby proto-oncogenes (https://www.science.org/content/article/gene-therapy-clinical-trial-halted-cancer-risk-surfaces).

7-       In the results part, you should describe your results, no reference should be cited. Explanation and comparison with other works must be in the discussion part. In the result part 3.1 it is enough if you start with “To evaluate whether the extra….”. From line 269-281 you can delete it.

[Reply] Done. In line with the present comment, we have reduced the use of references throughout the Results section. However, not every minor concept employed for our experimentation could be accommodated in the context of the Introduction without turning into a distraction there, and at times intermediate conclusions needed to be drawn in the Results section to justify subsequent experimental steps. Given that many publications employ references in the Results section in the same fashion, we therefore hope that their reduction rather than elimination there in the present manuscript is acceptable.

In the specific instant of section 3.1, we have moved almost all explanatory remarks to the corresponding section in the Introduction.

8-     In part 3.2, no need to describe previous work, this you can make it in the discussion. You can start the part 3.2 with “Hypothesizing that silencing of the aberrantly….”

[Reply] Done. The surplus text has been merged with the corresponding introductory section.

9-       For part 3.2.2, the same like previously, no need to describe previous work.

[Reply] Done. Reference to previous work has been reduced to the rationale of the specific shRNAmiR design choices.

10-    The data availability statement should be changed, the authors generate many data, the row data should be available for reader on reasonable request.

[Reply] The Data Availability Statement has been amended accordingly: “The data that support the findings of this study are available upon request.”

11-    The authors don’t analyze the constitution of the blood cells, it will be of great interest, if they show that the transduction doesn’t influencing the constitution of blood cells, at less analyze the constitution of the lymphocyte subset. Could you explain, why you don’t do that?

[Reply] The backbones of the specific GLOBE and pLKO.1-TRC vectors (as initially cited in our Patsali et al. 2018 (PMID: 29700171) paper) and their expression cassettes have been widely analyzed in HSPCs and in all manner of lineages (including embryonic stem cells), respectively, including for GLOBE in clonogenic assays (with scoring of CFUe, BFUe, CFU-GM, CFU-GEMM) for retrospective evaluation of oligopotency and proliferation potential. At the level of in vitro analyses, duplication of those efforts therefore seemed wasteful, in particular because once those vectors are analyzed in HSPCs from mobilized peripheral blood (mPB) for long-term repopulation potential in immunodeficient mice, those data will be more representative of vector performance (regarding both differentiation and proliferation potential) in the clinical setting than would be analyses based on expanded HSPCs from naive PB as used in the present study.

Beyond those aforementioned analyses based on clonogenic assays in methylcellulose or transplantation in e.g. NSG or NBSGW mice, specific lineage differentiation in vitro, such as for B and T cell lineages, as the question may suggest, are highly specialized for laboratories that have set up those cell systems and are not performed as in vitro assays in preclinical testing for gene addition therapies targeting the erythroid lineage.

Comments on the Quality of English Language

The english quality is fine, it needs only minor editing

[Reply] Language and consistency of the use of abbreviations have been revised throughout.

Reviewer 3 Report

Comments and Suggestions for Authors

In their manuscript „Evaluation of mono- and bi-functional GLOBE-based vectors for therapy of β-thalassemia by HBBβAS3 gene addition and mutation-specific RNA interference“ Koniali et al conduct a complex study on transfection of cell using GLOBE based vectors to determine the following expression of Hba among others in erythroid differentiating cells.

The manuscript clearly describes the multi-stage transfection study, precisely presenting the complexity of the methodology in each case. The characters are clear, although complex. The labels are clear and easy to read. The promoter study was precisely planned and rigorously implemented.

I strongly recommend this manuscript for publication.

Comments on the Quality of English Language

Minor editing can be performed.

Author Response

In their manuscript „Evaluation of mono- and bi-functional GLOBE-based vectors for therapy of β-thalassemia by HBBβAS3 gene addition and mutation-specific RNA interference“ Koniali et al conduct a complex study on transfection of cell using GLOBE based vectors to determine the following expression of Hba among others in erythroid differentiating cells.

The manuscript clearly describes the multi-stage transfection study, precisely presenting the complexity of the methodology in each case. The characters are clear, although complex. The labels are clear and easy to read. The promoter study was precisely planned and rigorously implemented.

I strongly recommend this manuscript for publication.

[Reply] We are thankful for the Reviewer’s positive assessment.

Comments on the Quality of English Language

Minor editing can be performed.

[Reply] Language and consistency of the use of abbreviations have been revised throughout.

Round 2

Reviewer 2 Report

Comments and Suggestions for Authors

1-     Adding only three line in the introduction about hemoglobinopathies and thalassemia is very short, at less add the types, distribution and incidence.

2-     To the response 3, you write in the original version, that you got the cells from ATCC, in this case you must mention the exact cell line. If the cells are self-generated, you mention the cells which are initially used.

3-     For response 11, it well be interesting for the reader if you mention that in the manuscript.

4-     In some part, the text format in the manuscript is not uniform

Author Response

1-     Adding only three line in the introduction about hemoglobinopathies and thalassemia is very short, at less add the types, distribution and incidence.

[Reply] With our eye on ATMP development, we had kept the basics of hemoglobinopathies brief, so as not to overburden the Introduction. However, we concede that the non-specialist audience of MDPI Cells would benefit from the indicated additional information. We have now expanded the first paragraph of the Introduction accordingly:

“Hemoglobinopathies are amongst the commonest monogenic disorders, are almost universally of recessive inheritance and cause clinical symptoms when affecting the α- and β-globin chains as constituents of the main adult hemoglobin, HbA (α2β2) [1]. Hemoglobinopathies may be brought about by toxic protein variants, such as by the sickling β-globin E6V amino acid change (βS), or by reduction or elimination of α- and β-globin expression in α- and β-thalassemia, respectively. Both sickle cell disease and β-thalassemia may reach extreme severity from infancy onwards, and without adequate management by blood transfusion and iron chelation, manifest as potentially lethal hemolytic anemias. Originally mostly confined to malaria regions, where carrier status for hemoglobinopathies confered a selective advantage by resistance to Plasmodium falciparum, hemoglobinopathies though considered rare diseases are now widespread through carrier migration and represent a global health challenge [2]. The last four decades have therefore seen tremendous efforts to develop advanced therapies for their treatment.”

2-     To the response 3, you write in the original version, that you got the cells from ATCC, in this case you must mention the exact cell line. If the cells are self-generated, you mention the cells which are initially used.

[Reply] With this second remark and to our embarrassment, the Reviewer has made us aware of an erroneous indication of (wt) MEL cell origin in our 2015 duplex PCR publication (PMID 26202078). While sound in its VCN measurement methodology, we have now removed reference to that paper in connection with MEL, MEL-HBB and MEL-HBBIVS cells here and have added the following passage in explanation instead:

“Murine erythroleukemia (MEL) cells of the APRT- cell line C88 [37], originally described by Deisseroth and Hendrick [38], and MEL derivatives expressing green fluorescent protein (MEL-GFP) [39] and the human-HBBIVSI−110(G>A)–transgenic (MEL-HBBIVS) cells [22] mimicking the HBBIVSI−110(G>A) splice defect, were used in the current study.”

The C88 MEL cells are not available through ATCC but since their creation and publication in Cell in 1978 (PMID 279411) have been used in dozens of publications, including our own of the creation of MEL-HBBIVS cells (PMID 29700171) and including the reutilization of our MEL-HBBIVS cells by others (PMID 33440169). The wild-type cells are referred to in patents (https://patents.google.com/patent/US5543319A/en) and publications (PMID 8812856, PMID 21416373) simply as “MEL (APRT-)” or “MEL-C88“ cells. Among our co-authors, the cells first arrived in Prof Michael Antonious’ laboratory through Frank Grosveld from the USA in around 1986 and have since been used continuously for vector characterization and as an erythroid murine model. As we have done in the past, we will be happy to pass those cells on to third parties upon request as research tools for the community.

3-     For response 11, it well be interesting for the reader if you mention that in the manuscript.

[Reply] We have now added a corresponding text passage in the Discussion in explanation of our choice of omitting certain preclinical analyses:

“Analyses undertaken here were restricted to those adding further information for vectors with well-characterized behavior and as appropriate for the cell substrates employed. Backbones of vectors GLOBE [34] and pLKO.1-TRC [58] and their expression cassettes have been widely analyzed in human cells, including long-term repopulating HSCs [59] and embryonic stem cells [22], respectively, and in preparation for clinical trials of the GLOBE vector [5], have also been analyzed in clonogenic assays for retrospective evaluation of oligopotency and proliferation potential [59]. Investigation of potential changes of vector properties, such as integration patterns and influence on differentiation and proliferation potential, in connection with the alternative inserts applied here, will be of critical importance before any clinical application, but would not be faithfully represented by the MEL cells or the expanded CD34+ cells employed in the present study. Such analyses and additional long-term repopulation and lineage analyses after transplantation in immunodeficient mice will be restricted to mobilized or bone-marrow aspirated cell samples once they become available. Instead, we focused here on transduction efficiencies and assessment of phenotype correction in the erythroid lineage at the mRNA, protein and cell-morphology level.”

4-     In some part, the text format in the manuscript is not uniform

[Reply] Prompted by this comment we have double-checked correct application of paragraph styles provided by the MDPI Cells template throughout, as well as corresponding font sizes and indents. In the process we noticed and corrected in the Results section three headings that instead of heading level 3 should have been heading level 2. Grateful to the Reviewer for having prompted this correction, we are nevertheless uncertain whether this correction addresses the more general concern about non-uniform text format. If the concern remains, we would be grateful for a more specific indication of non-uniformity or, in the end, for the vigilance and assistance of the MDPI print production team in its correction.